# Relation between Noise Pollution and Life Satisfaction Based on the 2019 Chinese Social Survey

**DOI:** 10.3390/ijerph19127015

**Published:** 2022-06-08

**Authors:** Dongliang Yang, Xiangheng Liu, Zhichao Ren, Mingna Li

**Affiliations:** 1Northeast Asian Research Center, Jilin University, Changchun 130012, China; yangdl@jlu.edu.cn; 2Department of Regional Economics, School of Northeast Asian, Jilin University, Changchun 130012, China; h924282818@163.com (X.L.); renzhichao654@163.com (Z.R.); 3Department of Chinese as a Foreign Language, School of Literature, Changchun University, Changchun 130000, China

**Keywords:** noise pollution, well-being, life satisfaction, living environmental satisfaction, education

## Abstract

Noise pollution is a leading cause of decreasing well-being of residents in both developed and developing countries. Improving residents’ well-being measured by life satisfaction is a key goal of government policy. Individuals with high life satisfaction usually have positive emotions, life orientation, and codes of conduct, which are positive and beneficial for individuals, families, and society as a whole. In order to supplement relevant research and provide policy suggestions for individuals, government, and societies, this study explores the relationship between noise pollution and the life satisfaction of Chinese residents. Based on data from 4869 observations from the Chinese Social Survey in 2019, the effect of noise pollution on life satisfaction is estimated by using ordinary least squares and propensity score matching methods. The results show that noise pollution has a significant negative effect on Chinese life satisfaction. Moreover, the effect is heterogeneous depending on individuals’ education levels and ages. Finally, residents’ living environment satisfaction is shown to be the potential mechanism by which noise pollution affects life satisfaction.

## 1. Introduction

Noise is a common phenomenon in daily life. Construction, industrial production, entertainment, and traffic are the main sources of noise. Noise pollution refers to the phenomenon that the generated noise exceeds the national environmental standards and interrupts normal life, and the work and study of others. Environmental noise pollution is a kind of energy pollution, which is a public hazard endangering human well-being. According to the Occupational Safety and Health Agency of America, the maximum exposure time of labored noise at 85 dB is 8 h, and the time at 110 dB is 89 s. Noise pollution can harm people’s nervous system and auditory systems and lead to symptoms such as memory loss and insomnia [1,2]. Many studies find the significant effect of noise pollution on cardiovascular disease, hearing loss, hyperactivity, inattention, and so on [2,3,4,5,6]. Noise pollution has become the second most damaging environmental factor to human health after air pollution [7,8]. Noise pollution can cause sleep disorders, anxiety, depression, aggravation of personal troubles, damage to interpersonal relationships, and other factors related to well-being and life satisfaction [9,10,11,12,13,14,15]. Undoubtedly, noise pollution threatens people’s health and quality of life. However, little attention has been paid to the role of noise pollution in affecting Chinese life satisfaction.

Improving the well-being of all residents is a critical goal of government policy. Life satisfaction is the reflection of an overall assessment of one person’s life [16,17]. Sato mentioned that the United States, Australia, and New Zealand consider life satisfaction as the core indicator of people’s happiness and well-being [18]. Individual life satisfaction is closely related to happiness. The pursuit of happiness and the realization of a “satisfactory life” are also considered to be the basic missions of psychology [19,20,21]. Residents’ life satisfaction is closely related to their daily work and life and affects their work efficiency and life quality. Many studies also indicate that dissatisfaction with life is harmful to individuals and society [22,23]. For example, feeling anxious and depressed about life can lead to a reduction in the perception of regulating negative emotions [24]. Serious despair in life can even lead to suicide and endanger society [25,26]. Therefore, exploring the reasons for undermining life satisfaction is essential for the well-being of individuals, families, and society.

Given the importance of life satisfaction, the determinants of life satisfaction are important issues. Many studies focus on the effect of economic changes and different personal characteristics on life satisfaction. For example, income and unemployment can affect a person’s life satisfaction, and the level of life satisfaction is different according to age, gender, and disposition [17,27,28]. In addition to the above aspects, environmental pollution as an external shock can also affect life satisfaction. Recent trends have caused a surge in research on the relationship between environment and individual subjective feelings [29,30,31].

Environmental concerns are some of the most urgent problems in the 21st century. Specifically, noise pollution can affect a human’s well-being and overall quality of life [7,31]. Some studies investigated the possible effect of specific noise pollution on life satisfaction. For example, Hegewald et al. found that individuals exposed to aircraft noise were at high risk of depression [11]. Jan and Vojtěch found that the effect of traffic noise on life satisfaction was significantly negative [32]. Lercher and Kofler found that traffic noise has a significant negative effect on people’s quality of life in mountainous rural areas [33]. Apart from the aforementioned studies, air pollution as a crucial factor can also reduce people’s subjective well-being [34,35,36]. Although the existing literature has revealed the determinants of life satisfaction from different aspects, there are few studies on the causal effect of noise pollution on life satisfaction.

This paper aims to fill the gap in the relation between noise pollution and Chinese life satisfaction. We analyze the effect of noise pollution on Chinese life satisfaction by using the data from the Chinese Social Survey (CSS) in 2019. We focus on this cohort for three reasons. First, with the rapid development of China’s economy and the continuous improvement of residents’ material life, the option of sacrificing the environment in pursuit of a high income is gradually being re-evaluated. Meanwhile, the awareness of environmental protection and pollution control is becoming stronger and stronger. Noise pollution represents one of the most pressing concerns for well-being and life quality. Second, considering the denser population in China, the harm of noise pollution is greater. As the largest developing country, China has generated serious noise pollution in the process of rapid urbanization [37]. Third, noise pollution can affect workers’ daily mood and life satisfaction, which is harmful to their work efficiency.

The novelty of this paper is four-fold. First, to the best of our knowledge, this is the first paper to analyze the relation between noise pollution and life satisfaction for the case of a developing country: China. The result and policy enlightenment is very important for improving the well-being of Chinese residents. Previous studies are mostly based on developed countries such as Britain and Germany [2,5,11]. Furthermore, our study investigated the general effect of noise pollution as an external shock on Chinese life satisfaction. Previous studies mainly focused on specific noise, such as traffic noise or construction noise [9,38,39]. Second, we address the effect by the propensity score matching (PSM) method. Third, we test the heterogeneous effect of noise pollution on life satisfaction by splitting the sample into different education levels and ages. These empirical results are beneficial for understanding the different effects of noise pollution on life satisfaction. Finally, this paper reveals the mechanism of noise pollution on life satisfaction through its effect on living environment satisfaction.

The remainder of this paper is structured as follows. In Section 2, we describe the study design, statistical analysis, data, and methodology. The empirical analysis is given in Section 3. In Section 4, we discuss the results and put forward policy implications. Conclusions are drawn in Section 5.

## 2. Materials and Methods

### 2.1. Study Design

This paper conducts a cross-sectional study by employing the data of the Chinese Social Survey (CSS) in 2019 [40], a nationwide, comprehensive, longitudinal survey in mainland China. The CSS project team collected data through face-to-face interviews. The sampling adopts stratified random probability sampling. Thirty-one provinces or their administrative equivalents were covered. The data contained 10,283 families from 151 cities and 604 villages in 2019. The exclusion of missing data yields 4869 usable observations. An econometric analysis using ordinary least squares (OLS) is conducted to examine the effect of noise pollution on life satisfaction. The study is approved by the Ethics Committee of Jilin University, and ethical clearance or equivalent approval to conduct the study was granted.

This paper not only studies the relationship between noise pollution and life satisfaction but also discusses the potential heterogeneity of the effects and the associated mechanisms. Hence, four hypotheses were proposed for our study.

Noise pollution can affect people’s life satisfaction from different aspects. For example, noise pollution can cause hearing loss [2] that may impair the quality of life. Due to the existence of noise pollution, depression, anxiety, stress, and other conditions may also be exacerbated [10,11], thus increasing dissatisfaction with life. Therefore, we propose Hypothesis 1.

**Hypothesis** **1** **(H1).**
*Noise pollution has a significant negative effect on residents’ life satisfaction.*


People with different levels of education often have different views and feelings on things and different attitudes towards life [41]. Compared with less-educated people, well-educated people are able to obtain a higher living standard that may also include a quieter environment. Those people with less education may be more tolerant of the negative effects of noise pollution but, in any case, may not be able to find an alternative, affordable location. The effect of noise pollution on life satisfaction may not be homogeneous for people at different education levels. Based on the above analysis, we propose Hypothesis 2.

**Hypothesis** **2** **(H2).**
*The effect of noise pollution on well-educated residents is stronger than their low-educated counterparts.*


Senile depression is very common in the world [42]. Many studies indicate the prevalence peak of depression is between the age of 55 and 74 years [43,44,45]. For the elderly population, the risk of diabetes, sleep difficulties, and other diseases increases. Many elderly people live alone due to the death of their spouse [46]. Therefore, compared with young people, these elderly groups suffer more pain from noise pollution. The effect of noise pollution on life satisfaction may vary depending on the person’s age. Therefore, we propose Hypothesis 3.

**Hypothesis** **3** **(H3).**
*Noise pollution has a greater negative effect on the life satisfaction of the elderly.*


Among the individuals we investigated, living environmental satisfaction represents residents’ subjective evaluation of their environmental situation in their daily life. Individuals with a high level of living environmental satisfaction usually have more positive emotions and life attitudes than the ones with a lower level [13]. Therefore, individuals with a high level of living environmental satisfaction may have greater life satisfaction [47,48] In addition, the existing literature indicates that noise pollution has a significant negative effect on living environmental satisfaction [31,39]. Therefore, we propose Hypothesis 4.

**Hypothesis** **4** **(H4).**
*Noise pollution has an effect on residents’ life satisfaction through its effects on living environmental satisfaction.*


### 2.2. Statistical Analysis

Using the econometric software Stata version 16 for statistical analysis, we report the mean, standard deviation, minimum and maximum of variables in Table 1. Given life satisfaction is a continuous variable; we used OLS to examine the relationship between noise pollution and life satisfaction. To deal with endogeneity problems, propensity score matching (PSM) was used. (We used the “psmatch2” package to calculate the effect of the various propensity score matching methods). To obtain robust results, we estimated the effect of noise pollution on life satisfaction again by using a probit model. To test the mechanism, we used OLS to estimate the effect of noise pollution on living environment satisfaction and the effect of living environment satisfaction on life satisfaction. All reported *p*-values were two-tailed.

### 2.3. Variables and Descriptive Statistics

The dependent variable in this study was self-evaluated life satisfaction. The CSS contained the question “Summed up, how satisfied are you with your life” and provided ten level responses from 1 to 10. The larger the value, the higher the respondent’s life satisfaction.

The independent variable was noise pollution. According to the question “Is the noise pollution serious in your location” in the CSS, we considered the answer of the respondent as a dummy variable, which was equal to 1 for a positive answer that indicated the respondent was suffering the noise pollution, otherwise, it was 0.

The control variables included gender, ethnicity, religion, age, education, income, social activities, and house property. Gender, ethnicity, religion, social activities, and house property were set as dummy variables. Education level was captured by a categorical variable. Income was constructed as an ordinal category variable according to its range. Age was a continuous variable. The descriptive statistics of the variables are reported in Table 1. In the sample, residents were 49-years-old on average, and 56.685% of them were female. The average resident’s life satisfaction score was 7.123. About 60% of residents felt noise pollution around them, and the average living environmental satisfaction score was 6.857.

### 2.4. Empirical Methodologies

The ordinary least square method was used to estimate the effect of noise pollution on residents’ life satisfaction, as follows:(1)$${\mathrm{lsatisfaction}}_{i}=\alpha_{0}+\beta_{0}{\mathrm{nopollution}}_{i}+{\lambda \mathrm{control}}_{i}+\varepsilon_{i}$$
where ${\mathrm{lsatisfaction}}_{i}$ represents the dependent variable (residents’ life satisfaction), ${\mathrm{nopollution}}_{i}$ represents the independent variable (for the evaluation of noise pollution in the residence, it is 1 if there is noise pollution, otherwise it is 0), ${\mathrm{control}}_{i}$ is a vector of observable determinants of residents’ life satisfaction. $\varepsilon_{i}$ is the error term, $\alpha_{0}$ is the constant term, and $\beta_{0}$ is the coefficient of the independent variable, which reflected the effect of noise pollution on residents’ life satisfaction.

Propensity score matching (PSM) was employed to deal with endogeneity problems. In this case, we regarded the noise pollution value of 1 as the exposure group and 0 as the control group. The core idea of utilizing PSM was to balance the observable characteristics between the exposure and control groups by matching the propensity score and calculating the effect of noise pollution on life satisfaction.
(2)$$\tau_{\mathrm{ATT}}=E[Y_{1i}+Y_{0i}|D_{i}=1]$$
where $Y_{1i}$ and $Y_{0i}$ are potential outcome variables, $D_{i}$ represents exposure or no-exposure condition.

## 3. Empirical Results

### 3.1. Baseline Results

When examining the relationship between noise pollution and life satisfaction, some variables may be highly correlated, such as income and education. Those correlations among variables usually cause concerns about multicollinearity, which may lead to bias in the estimation. We employed the variable inflation coefficient (VIF) and tolerance (1/VIF) to check the multicollinearity of the model. Table 2 reports the VIF and tolerance of each variable. The VIF of each variable was less than the rule-of-thumb value of 10, that is, the tolerance was more than 0.1, indicating that multicollinearity was not a major matter in our model.

Table 3 reports the baseline result on the effect of noise pollution on life satisfaction. The regression coefficient of noise pollution was −0.329 (95% confidence interval: −0.462−0.196), *p* < 0.01. OLS regression results showed that noise pollution has a significant negative effect on life satisfaction, and the results verify Hypothesis 1. In addition, the coefficients of Sex, Age, Education, Income, and Social activities were positive and statistically significant. Some conclusions can be drawn. Specifically, the life satisfaction of women was higher than that of men. The higher the education level, the more satisfied people are with life. Participating in social activities significantly promotes life satisfaction. Residents who own a house are more satisfied with their life.

### 3.2. Endogeneity

Considering the selection bias, we used the propensity score matching (PSM) technique to estimate the effect of noise pollution on life satisfaction. We checked the covariate balance of the exposure and control groups. Here, two methods were used. The first one was to compare the means of covariates in two groups. Table 4 reports the averages for the two groups, and *t*-statistics and *p*-values assess the test of the null hypothesis of equality of means of the covariates in the exposure and control groups. It is clear that the two groups differed substantially in the distribution of their background characteristics before matching. The subsample of individuals exposed to noise pollution had, on average, a higher education level, income, and more social activities. Such differences may lead to the bias of the OLS result. After matching, we found the matched *p*-values were larger than 0.1 in most of the cases, indicating that the covariate balance of the two groups was greatly improved. The second one was to compute the standardized bias. Figure 1 reports the changes in standardization bias of the eight covariates after matching respectively. All the standardized deviations between the exposure and control groups were less than 5% after matching, indicating that there was no significant difference between the two groups. Thus, the covariates of the exposure and control groups were balanced.

According to Heckman et al. [49], it is necessary to access the overlap and region of common support between the exposure group and the control group in the application of PSM. Figure 2 reports the kernel density distribution of the linearized propensity score for the exposure and control groups after matching. We found that the curve of the exposure group (red) and the curve of the control group (blue) were similar including the value range of variables on the *X*-axis, peak value, and shape of the curve, providing evidence that the two groups had good overlap and region of common support.

In this paper, several matching methods were used to estimate the effects after propensity score matching, including nearest neighbor matching, radius matching, local linear matching, and kernel matching. According to the results in Table 5, all the effects of various matching methods were significantly negative, proving that noise pollution has a negative effect on an individual’s life satisfaction.

### 3.3. Robustness Check

To confirm the reliability of the results, a robustness check was conducted. We used a probit model to estimate the effect of noise pollution on life satisfaction. We used a dummy variable to measure life satisfaction. If the value of life satisfaction was more than 7.123, the dummy variable was equal to1, otherwise, 0. Table 6 presents the estimated results.

The regression coefficient of noise pollution was −0.186 (95% confidence interval: −0.262 −0.111), *p* < 0.01, indicating that noise pollution has a significant negative effect on life satisfaction. The marginal effect of noise pollution was −0.073 (95% confidence interval: −0.103−0.044), *p* < 0.01. In conclusion, the estimated results were consistent with the results in the previous section, which further confirmed Hypothesis 1.

### 3.4. Heterogeneity

Regarding the different effects of noise pollution on life satisfaction, we investigated the heterogeneity of effects by splitting the sample. First, we divided the sample into two groups: well-educated and less-educated individuals. The results are shown in Table 7. The coefficients of noise pollution were −0.435 and −0.264 respectively, indicating that people with high education have a stronger response to noise pollution than those with low education. Hypothesis 2 is verified.

Second, we divided the samples into two groups by age. One group was made up of older persons whose age was more than 50 years old and the rest belonged to the other group. The results are shown in Table 8. The coefficients of noise pollution were −0.265 and −0.371 respectively, indicating that noise pollution has a greater negative effect on older persons. Hypothesis 3 was verified.

### 3.5. Mechanisms

To better understand the relationship between noise pollution and life satisfaction, we examined the mechanism by which noise pollution affects residents’ life satisfaction. In order to verify Hypothesis 4, we considered the living environmental satisfaction as the mechanism variable; with this variable ranging from 1 to 10, where 1 meant very dissatisfied with the environment and 10 meant very satisfied with the environment. Table 9 reports the result. The coefficient of noise pollution on living environment satisfaction was −1.257 (95% confidence interval: −1.391 −1.124), *p* < 0.01. It showed that noise pollution has a significant negative effect on residents’ satisfaction with their living environment. The coefficient of the residential environment on life satisfaction was 0.285 (95% confidence interval: 0.253 0.316), *p* < 0.01, which indicated that the living environment satisfaction has a significant positive effect on life satisfaction. It is obvious that noise pollution has a negative effect on residents’ life satisfaction by harming their living environment satisfaction. Therefore, the results verified Hypothesis 4.

## 4. Discussion

Some studies are related to the theme of this paper, however, they focus on a specific type of noise pollution. For instance, Jan and Vojtěch explored the effect of railway and highway traffic noise on living environment satisfaction and life satisfaction by capturing noise exposure values and found that traffic noise has a significant negative effect on environmental satisfaction but has no significant effect on overall life satisfaction [32]. Pedersen and Botteldooren used questionnaires to investigate people’s evaluation of traffic noise pollution in the community and found that people are generally annoyed by traffic noise and think it has affected their quality of life [38,39]. Ma et al. investigated the correlation between residents’ evaluation of construction noise and traffic noise in Beijing and urban residents’ mental health symptoms, and the results showed that both of them are significantly correlated [13]; Xiao et al. found that construction noise may bring significant health risks to nearby residential communities [9]. However, the above papers do not provide the heterogeneity and mechanisms of the analysis. Our study not only investigates the effect of noise pollution on residents’ life satisfaction but it also analyzes the heterogeneity and mechanism.

Although there is mounting evidence indicating that noise (such as traffic noise) may have a negative effect on individuals’ health, few findings concern the effect of one’s subjective evaluation on life satisfaction. Furthermore, our study also analyzes the different effects of noise pollution on a different sample of people. The empirical results support that the effect of noise pollution on people’s life satisfaction is heterogeneous depending on education level and age. That is, well-educated people have a stronger response to noise pollution than those with lower education levels. Older people respond more strongly to noise pollution. Finally, this paper finds one mechanism of noise pollution affecting people’s life satisfaction.

However, this paper has some limitations. First, we estimated the short-term effect of noise pollution on residents’ life satisfaction. Unfortunately, due to the cross-sectional data constraints, we fail to consider the long-term effect of noise pollution on their life satisfaction. Second, we measure the effect of noise pollution on life satisfaction by investigating residents’ subjective evaluation of noise pollution, rather than conducting a physical experiment. Third, although we have added many control variables that affect people’s life satisfaction, there may be some omitted variables that are unobserved or difficult to measure. It is difficult to estimate the causal relationships between noise pollution and life satisfaction. One future research direction would be to study the long-term effect of noise pollution on life satisfaction by using panel data and including more affecting factors as control variables. In addition, we can also study the effect of different types of noise pollution on life satisfaction.

Several policy implications can be derived from our research. First, our study suggests that noise pollution has a notable adverse effect on residents’ life satisfaction. Therefore, the government should take action to strengthen the control of noise from construction, traffic, entertainment, and others. The actions include collecting opinions of all the residents, holding hearings, legislation, and active implementation. Specifically, it is necessary to set up noise barriers to reduce the noise of expressways, railways, and road traffic in urban residential areas. The community should propose some reasonable rules to keep their entertainment sounds within a safe range. Some activities held in squares and parks near the residential area should comply with the noise emission regulation, such as decibel, time limitation, and so on. Moreover, the remodeling times in high-rise residential areas should be limited to avoid disturbance to the surrounding residents. Finally, the government should punish strictly noise-making behaviors such as motor vehicle “street bombing” and fine those who violate the noise management regulations.

## 5. Conclusions

In recent years, there is more and more research on people’s life satisfaction, but the investigation of its influencing factors is different. Using 4869 observations from the 2019 CSS data, this paper studies the relationship between noise pollution and the life satisfaction of Chinese residents. One of the most important findings is that noise pollution has a negative effect on life satisfaction.

To further analyze the heterogeneity, we split the samples into different education levels and different age groups. On the one hand, it shows that the effect of noise pollution on people with high education is stronger than that on people with a lower level of education. On the other hand, it shows that noise pollution has a stronger effect on older groups. Finally, this paper also studies the mechanism of noise pollution affecting life satisfaction, which shows that noise pollution affects life satisfaction by affecting residents’ satisfaction with the living environment.

## Figures and Tables

**Figure 1 ijerph-19-07015-f001:**
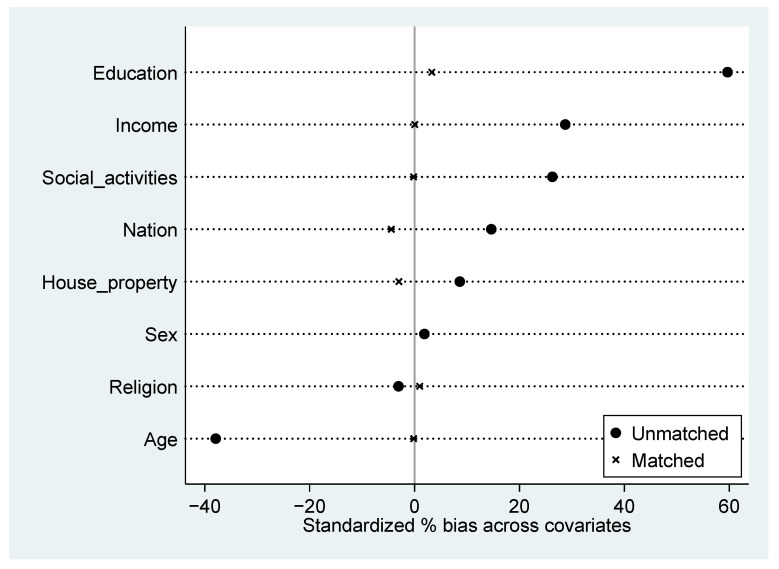
Standardized bias of each covariate before and after matching.

**Figure 2 ijerph-19-07015-f002:**
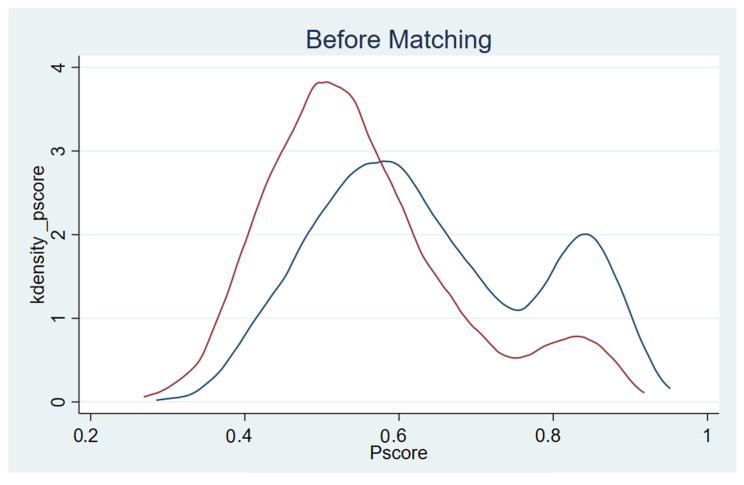
Kernel density distribution of propensity score (after matching).

**Table 1 ijerph-19-07015-t001:** Descriptive statistics of the variables.

Variable	Definition	Mean	SD	Min	Max
Life satisfaction	Chinese residents’ life satisfaction	7.123	2.207	1	10
Noise pollution	Evaluation of noise pollution in environment, 1 for existing, 0 for none	0.606	0.489	0	1
Sex	1 for female, 0 for male	0.567	0.496	0	1
Ethnicity	1 for Ethnic Han, 0 for others	0.918	0.274	0	1
Religion	1 for no religion, 0 for religion	0.862	0.345	0	1
Age	Individual’s age	49.42	14.171	21	72
Education	Education level	3.700	2.113	1	9
Income	Individual’s income (rank division)	2.130	0.964	1	7
Social activities	1 for attending, 0 for not	0.409	0.492	0	1
House property	1 for own, 0 for none	0.932	0.237	1	0
Environmental satisfaction	Residents’ living environmental satisfaction	6.857	2.295	1	10

**Table 2 ijerph-19-07015-t002:** The variance inflation factor of each variable.

Variable	VIF	1/VIF
Noise pollution	1.1	0.913
Sex	1.08	0.924
Ethnicity	1.04	0.963
Religion	1.04	0.966
Age	1.41	0.707
Education	1.69	0.593
Income	1.21	0.826
Social activities	1.18	0.846
House property	1	0.996
Mean VIF	1.19	

Note: VIF represents variable inflation factor.

**Table 3 ijerph-19-07015-t003:** Ordinary least square (OLS) results of the effect of noise pollution on residents’ life satisfaction.

Variable	Coefficient	95% CI
Noise pollution	−0.329 *** (0.068)	−0.462 −0.196
Sex	0.148 ** (0.065)	0.020 0.276
Ethnicity	0.068 (0.130)	−0.186 0.322
Religion	0.154 (0.096)	−0.034 0.341
Age	0.01 *** (0.003)	0.005 0.015
Education	0.134 *** (0.017)	0.100 0.168
Income	0.149 *** (0.033)	0.086 0.213
Social activities	0.289 *** (0.067)	0.158 0.420
House property	0.671 *** (0.146)	0.385 0.958
Constant	4.345 (0.273)	3.811 4.880

Note: *** *p* < 0.01, ** *p* < 0.05, Robust standard errors are reported in parentheses.

**Table 4 ijerph-19-07015-t004:** The balance test of covariates in exposure and control groups.

Variable	Matching Status	Mean		*t*-Statistic	*p*-Value
		Exposure	Control		
Sex	Before	0.571	0.561	0.64	0.523
	After	0.570	0.560	0.79	0.428
Ethnicity	Before	0.935	0.894	5.11	0.000
	After	0.935	0.947	−2.03	0.043
Religion	Before	0.858	0.868	−1.04	0.298
	After	0.857	0.854	0.37	0.713
Age	Before	47.356	52.590	−12.8	0.000
	After	47.371	47.397	−0.07	0.945
Education	Before	4.166	2.983	19.86	0.000
	After	4.162	4.097	1.13	0.259
Income	Before	2.236	1.967	9.61	0.000
	After	2.233	2.233	0.01	0.996
Social activities	Before	0.459	0.332	8.9	0.000
	After	0.459	0.460	−0.07	0.942
House property	Before	0.932	0.953	2.89	0.004
	After	0.932	0.926	−1.05	0.295

**Table 5 ijerph-19-07015-t005:** Propensity score matching (PSM) analysis of the effect of noise pollution on residents’ life satisfaction.

Variable	Nearest Neighbor Matching (k = 4)	Radius Matching	Local Liner Matching	Kernel Matching
Noise pollution	−0.297 *** (0.085)	−0.340 *** (0.077)	−0.335 *** (0.108)	−0.343 *** (0.076)

Note: *** *p* < 0.01.

**Table 6 ijerph-19-07015-t006:** Probit results of the effect of noise pollution on residents’ life satisfaction.

Variable	Coefficient	95% CI	Marginal Effect	95% CI
Noise pollution	−0.186 *** (0.039)	−0.262 −0.111	−0.073 *** (0.015)	−0.103 −0.044
Sex	0.044 (0.038)	−0.031 0.118	0.017 (0.015)	−0.012 0.046
Ethnicity	−0.021 (0.068)	−0.153 0.112	−0.008 (0.027)	−0.060 0.044
Religion	0.024 (0.053)	−0.081 0.129	0.009 (0.021)	−0.032 0.051
Age	0.006 *** (0.002)	0.003 0.009	0.002 *** (0.001)	0.001 0.003
Education	0.061 *** (0.011)	0.039 0.083	0.024 *** (0.004)	0.015 0.032
Income	0.069 *** (0.021)	0.028 0.110	0.027 *** (0.008)	0.011 0.043
Socialactivities	0.130 *** (0.040)	0.052 0.208	0.051 *** (0.016)	0.020 0.081
Houseproperty	0.282 *** (0.078)	0.130 0.435	0.111 *** (0.030)	0.051 0.170
Constant	−0.344 (0.153)	−0.643 −0.044		

Notes: *** *p* < 0.01.

**Table 7 ijerph-19-07015-t007:** OLS results of the heterogeneous effect of noise pollution by education level.

Variable	Well-Educated	Less-Educated
Noise pollution	−0.435 *** (0.097)	−0.264 *** (0.087)
Sex	0.056 (0.084)	0.243 ** (0.094)
Ethnicity	−0.084 (0.179)	0.091 (0.165)
Religion	0.347 ** (0.134)	0.091 (0.128)
Age	−0.002 (0.003)	0.017 *** (0.004)
Income	0.109 *** (0.038)	0.291 *** (0.056)
Social activities	0.274 *** (0.090)	0.254 *** (0.095)
House property	0.491 ** (0.192)	0.761 *** (0.199)
Constant	7.744 *** (0.343)	5.988 *** (0.376)
Observations	1806	3063

Notes: *** *p* < 0.01, ** *p* < 0.05.

**Table 8 ijerph-19-07015-t008:** OLS results of the heterogeneous effect of noise pollution by different ages.

Variable	Age ≤ 50	50 > Age
Noise pollution	−0.265 ** (0.094)	−0.371 *** (0.096)
Sex	0.205 ** (0.087)	0.104 (0.097)
Ethnicity	0.104 (0.164)	0.020 (0.205)
Religion	0.065 (0.126)	0.232 (0.142)
Education	0.138 *** (0.018)	0.067 ** (0.034)
Income	0.053 (0.037)	0.392 *** (0.062)
Social activities	0.271 *** (0.086)	0.305*** (0.103)
House property	0.683 (0.175)	0.633 *** (0.242)
Constant	6.872 *** (0.288)	6.645 *** (0.363)
Observations	2341	2528

Note: *** *p* < 0.01, ** *p* < 0.05.

**Table 9 ijerph-19-07015-t009:** OLS results of the mechanism of noise pollution affecting life satisfaction.

	Living Environment Satisfaction	Life Satisfaction
Noise pollution	−1.257 *** (0.068)	
Residential environment		0.285 *** (0.016)
Sex	−0.096 (0.066)	0.176 *** (0.062)
Ethnicity	−0.261 ** (0.116)	0.145 (0.122)
Religion	−0.075 (0.092)	0.174 * (0.092)
Age	0.000 (0.003)	0.010 *** (0.002)
Education	0.068 *** (0.018)	0.116 *** (0.016)
Income	0.004 (0.034)	0.149 *** (0.032)
Social activities	0.183 *** (0.069)	0.237 *** (0.064)
Houseproperty	−0.384 *** (0.145)	−0.560 *** (0.141)
Constant	8.060 *** (0.265)	4.059 *** (0.281)

Note: *** *p* < 0.01, ** *p* < 0.05, * *p* < 0.10.

## Data Availability

The CSS data can be accessed through its official website (http://csqr.cass.cn/DataExplore/?ProjectID=2018061909463245927261066314 (accessed on 21 April 2022)).

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
