# Peer review of "Relation between Noise Pollution and Life Satisfaction Based on the 2019 Chinese Social Survey"

_ijerph, 2022, doi:10.3390/ijerph19127015_

Round 1

Reviewer 1 Report

Dear Author,

the revised version have adressed the reviewer´s comments well.

One comment: The citation after line 20 has to show the changed title

Kind regards

Author Response

Thanks. I corrected. The title is “Relation between Noise Pollution and Life Satisfaction Based on the 2019 Chinese Social Survey”.

Reviewer 2 Report

Thank you for responding to my inquiries.

Author Response

Thanks.

Reviewer 3 Report

The authors have improved all the remarks of the reviewer.

Author Response

Thanks.

This manuscript is a resubmission of an earlier submission. The following is a list of the peer review reports and author responses from that submission.

Round 1

Reviewer 1 Report

Dear Author,

The paper with the title Impact of Noise Pollution on Life Satisfaction: Based on the 2019 Chinese Social Survey definitely fits in the scope of the section Health-Related Quality of Life and Well-Being.

The noise pollution was derived via an item of the survey questionnaire. The study participants were asked whether there was noise pollution in the area or not. The answer is dichotomous. However, this is a perceived effect and not an objective noise exposure. This is also not an independent variable, since the perception of noise itself also depends on noise sensitivity and other personal influencing factors. By asking, "Is the noise pollution serious in your location" the noise pollution is already rated by the respondents. The state of the art for determining sound exposure in cross-sectional studies is calculation.

True noise exposure is also not dichotomous. It varies on the strength and type of sources, the distance to the receiver and others, and is a metric measure.

I don´t agree that by use of a perceived noise pollution, the research questions can be answered in the specific terms. However, the method of propensity score matching (PSM) appears to be a promising method for controlling confounders.

If the 2019 Chinese Social Survey data should be used for this additional research, the noise pollution would have to be interpreted in another way. The relation between perceived noise pollution and life satisfaction can be investigated. However, this means a clear intervention in the structure of the manuscript including the title.

Kind regards

Author Response

Thank you for the opportunity to revise our manuscript entitled “Impact of Noise Pollution on Life Satisfaction: Based on the 2019 Chinese Social Survey” (ijerph-1717181). We also thank for the your valuable and insightful comments that help us substantially improve our paper. We have studied the comments carefully.

As reviewer pointed out, there are two methods. In section 4, we simply discuss the problem on the measurement of noise pollution. Some researcher used objective indicator, such as traffic noise, for instance, Jan and Vojtěch [33]. Some used questionnaires to investigate people's evaluation of noise pollution, for instance, Pedersen [38]and Botteldooren[39]. There are advantage and disadvantage for two methods. In our study, I choose the evaluation method. One reason is data limitation. Another reason is to test the casual effect of noise evaluation of different people, even if they have different perception standard.

I think the title could be changed to “Impact of Evaluation of Noise Pollution on Life Satisfaction: Based on the 2019 Chinese Social Survey” if necessary. I am not sure which one is better. The title of Botteldooren’s paper is “The Influence of Traffic Noise on ………”.

Reviewer 2 Report

I have read the article entitled "Impact of Noise Pollution in Life Satisfaction: BAsed on the 2019 Chinese Social Survey", which really present relevant information about the relationship between noise perception and life satisfaction. 

I just have few comments to the authors, as follows:

1.- Introduction section:

Authors mentioned several sources of noise (not in singular in line 26). OSHA is of America (not American, line 30). This regulation is about laboral noise.  Please include this remark for avoiding confusion  (line 31).

Please include any relevant reference about memory loss and insomnia effects (line 33). And also for the 2/3 of chinese urban residents living in an environment  with excessive noise (line 36). The statement: "recent trends have caused a surge in research on the relationship between environment and individual subjective feelings", need a relevant quote (line 62). 

In line 86, please consider the reference: https://www.sciencedirect.com/science/article/abs/pii/S0003682X18303608 for the sentence: "China is experiencing serious noise pollution in the high speed urbanization. 

IN line 94, plesase change UK instead of Britain. 

2.- Materials and Methods.

There is a typo error in line 116 (10283in)

In table 1 the minimum life satisfaction is 0, but in line 172 the level responses range from 1 (not 0).

3.- Empirical Results

In table 2, what is the contribution of 1/VIF. What does it mean? Can you provide any argument about that?

Please include a deeper interpretation of figures 2 and 3. Also about the marginal effect obtained in table 6

In line 295 the living environment satisfaction ranges from 1 to 10, but next 0 (not 1) means very very dissatisfied....

4.- Discussion section

There is a typo error in line 327 (supportthat)

General revision: cites in the text (not numbers, just authors) should be revised (some of them include first names, instead of et al) i.e.: lines: 67, 68, 313, 315, 318.

Author Response

Thank for the reviewer’ valuable and insightful comments that help us substantially improve our paper. We have studied the comments carefully and converted them into ten question (Q1-Q10). We made corresponding corrections as follows. The detailedcorrections are enlosed too. To show the revisions more clearly, we have marked all corresponding revisions in the revised manuscript. 

1.- Introduction section:

Q1. Authors mentioned several sources of noise (not in singular in line 26). OSHA is of America (not American, line 30). This regulation is about laboral noise.  Please include this remark for avoiding confusion  (line 31).

Response 1: I corrected. The corrected is highlighted as below. “According to Occupational Safety and Health Agency of America, the maximum exposure time of labored noise at 85 dB is 8 hours,”

Q2. Please include any relevant reference about memory loss and insomnia effects (line 33). And also for the 2/3 of chinese urban residents living in an environment with excessive noise (line 36). The statement: "recent trends have caused a surge in research on the relationship between environment and individual subjective feelings", need a relevant quote (line 62).

In line 86, please consider the reference: https://www.sciencedirect.com/science/article/abs/pii/S0003682X18303608 for the sentence: "China is experiencing serious noise pollution in the high speed urbanization.

Response 2: I corrected. I deleted “2/3 of Chinese urban residents living in an environment with excessive noise (line 36)” due to no authority literature stated clearly. I added some new literatures and adjusted the order of them. The corrected is highlighted as below.

“Noise pollution can harm people's nervous system and auditory system, and lead to symptoms such as memory loss and insomnia [1-2].”

“Recent trends have caused a surge in research on the relationship between environment and individual subjective feelings [30-31].”

“China is experiencing serious noise pollution in the high speed urbanization [32]. ”

[32] Di, H.; Liu, X.; Zhang, J.; Tong, Z.; Ji, M.; Li, F.; Feng, T.; Ma, Q. Estimation of the Quality of an Urban Acoustic Environment Based on Traffic Noise Evaluation Models. Applied Acoustics. 2018,141, 115-124. https://doi.org/10.1016/j.apacoust.2018.07.010.

Q3. IN line 94, plesase change UK instead of Britain.

Response 3: I corrected. The corrected is highlighted as below. “Previous studies are mostly based on developed countries such as the UK and Germany.”

2.- Materials and Methods.

Q4. There is a typo error in line 116 (10283in)

Response 4: The annual survey begun at 2005. About 7000 to 10000 families from 151 cities and 604 villages were investigated every year. In 2019, the number of families is 10283, a little bigger than 10000. The corrected is highlighted as below. “The data contained 10283 families from 151 cities and 604 villages in 2019.”

Q5. In table 1 the minimum life satisfaction is 0, but in line 172 the level responses range from 1 (not 0).

Response 5: I corrected. In table 1 the minimum life satisfaction is 1.

3.- Empirical Results

Q6. In table 2, what is the contribution of 1/VIF. What does it mean? Can you provide any argument about that?

Response 6: Both VIF and 1/VIF are statistical indicator. They are reciprocal. 1/VIF is named as tolerance. The corrected is highlighted as below. We employ the variable inflation coefficient (VIF) and tolerance (1/VIF) to check multicollinearity of model. Table 2 reports the VIF and tolerance of each variable. The VIF of each variable is less than the rule-of-thumb value of 10, that is, the tolerance is more than 0.1, indicating that multicollinearity is not a major trouble in our model.

Q7. Please include a deeper interpretation of figures 2 and 3. Also about the marginal effect obtained in table 6

Response 7: Another essential condition for implementing matching is to meet the common support assumption. To test the assumption, we draw Kernel density distribution of the propensity score of the treatment group and the control group after PSM (Figure 2).

The corrected is highlighted as below. “We report the kernel density distribution in the two groups in Figures 2. The propensity score distributions of the two groups are similar after matching, providing evidence that the two groups have the region of common support.”

Q8. In line 295 the living environment satisfaction ranges from 1 to 10, but next 0 (not 1) means very very dissatisfied....

Response 8: I have corrected this mistaken. The corrected is highlighted as below.  “which ranges from 1 to 10, where 1 means very dissatisfied with the environment and 10 means very satisfied with the environment.”

4.- Discussion section

Q9. There is a typo error in line 327 (supportthat)

Response 9: I corrected.

Q10. General revision: cites in the text (not numbers, just authors) should be revised (some of them include first names, instead of et al) i.e.: lines: 67, 68, 313, 315, 318.

Response 10: I corrected.

Reviewer 3 Report

Thank you for submitting this article for publication.  I must admit that in over 40 years of performing peer reviews for journals, I have never run into a manuscript that was ready for publication as is.

You may want to clarify in the introduction that whereas noise levels in excess of 85 dBA can cause measurable hearing loss, in this article, you are referring to noise levels below 85 dBA. This was implied, but never stated explicitely.

The only issue I see with this article are the occasional comments that lower levels of noise can cause health effects.  After reviewing the literature myself on this topic, I agree that environmental noise can cause serious degradation to the quality of life, BUT the literature is still not conclusive to say that environmental noise can cause health effects.  There can be short term increases in stress and its associated hormones and blood chemistry and also short term effects on sleep and on school education, but we cannot (yet) conclude that this will result in long term health and/or educational issues.  That is, these studies tend to be only correlative. You may want to remove the occasional sentence or reference to permanent "health effects".

Another issue (strength), which I thank you for bringing to the forefront, is that most of the literature in large reviews (such as the 8 studies from WHO) are from American and Western European sources with very few from China.  This study fills this important gap.

Author Response

Thank for the reviewer’ valuable and insightful comments that help us substantially improve our paper. We have studied the comments carefully and made corresponding corrections. The point-by-point responses to the reviewer’s comments are as follows. To show the revisions more clearly, we have marked all corresponding revisions in the revised manuscript.

Q1. You may want to clarify in the introduction that whereas noise levels in excess of 85 dBA can cause measurable hearing loss, in this article, you are referring to noise levels below 85 dBA. This was implied, but never stated explicitely.

Response 1: 85 dBA is an important threshold for individual’s health. Therefore, I think the research on the noise pollution is meaningful. But it is difficult to make physical experiment of human to test the harmful effect of sound level. We study it by the survey questionnaire. It is true that no direct casual relationship between 85 dBA and objective of paper, I think it may bring out most people’s interests on noise pollution.   

Q2. The only issue I see with this article are the occasional comments that lower levels of noise can cause health effects.  After reviewing the literature myself on this topic, I agree that environmental noise can cause serious degradation to the quality of life, BUT the literature is still not conclusive to say that environmental noise can cause health effects.  There can be short term increases in stress and its associated hormones and blood chemistry and also short term effects on sleep and on school education, but we cannot (yet) conclude that this will result in long term health and/or educational issues.  That is, these studies tend to be only correlative. You may want to remove the occasional sentence or reference to permanent "health effects".

Response 2: As reviewer pointed out, environmental noise can cause serious degradation to the quality of life. When come to health, there is difficult to provide quantitative evidence to test the casual relationship between noise and health. Maybe the bad effect on mental health is credible. I adopt the suggestion to remove some sentences on health effects. One corrected place is highlighted as below. “Environmental problem is one of the most urgent problems in the 21st century. Specifically, Noise pollution can affect human's well-being and overall quality of life [7,31].”

Round 2

Reviewer 1 Report

Dear Author,

Your rebuttal convinced me but there is definitely a need to change the title to “Impact of Evaluation of Noise Pollution on Life Satisfaction: Based on the 2019 Chinese Social Survey”

Kind regards

Author Response

Thanks for your comment. We adopt your excellent advice. The tittle is changed as "Impact of Evaluation of Noise Pollution on Life Satisfaction: Based on the 2019 Chinese Social Survey". Thanks again.